# Approach to Knee Arthropathy through 180-Degree Immersive VR Movement Visualization in Adult Patients with Severe Hemophilia: A Pilot Study

**DOI:** 10.3390/jcm11206216

**Published:** 2022-10-21

**Authors:** Roberto Ucero-Lozano, Raúl Pérez-Llanes, José Antonio López-Pina, Rubén Cuesta-Barriuso

**Affiliations:** 1Department of Physiotherapy, European University of Madrid, 28670 Madrid, Spain; 2Department of Physiotherapy, Catholic University San Antonio-UCAM, 30107 Murcia, Spain; 3Department of Basic Phycology and Methodology, University of Murcia, 30100 Murcia, Spain; 4Department of Surgery and Medical-Surgical Specialties, University of Oviedo, 33006 Oviedo, Spain

**Keywords:** hemophilia, knee, virtual reality exposure therapy, joint pain, physiotherapy

## Abstract

(1) Background: Hemarthrosis is a typical clinical manifestation in patients with hemophilia. Its recurrence causes hemophilic arthropathy, characterized by chronic joint pain. Watching movement recorded from a first-person perspective and immersively can be effective in the management of chronic pain. The objective of this study was to evaluate the effectiveness of an immersive virtual reality intervention in improving the pain intensity, joint condition, muscle strength and range of motion in patients with hemophilic knee arthropathy. (2) Methods: Thirteen patients with hemophilic knee arthropathy were recruited. The patients wore virtual reality glasses and watched a flexion–extension movement of the knee on an immersive 180° video, recorded from a first-person perspective over a 28-day period. The primary variable was the pain intensity (visual analog scale). The secondary variables were the joint status (Hemophilia Joint Health Score), quadriceps and hamstring strength (dynamometry), and range of motion (goniometry). (3) Results: After the intervention period, statistically significant differences were observed in the intensity of the joint pain (Standard error [SE] = 19.31; 95% interval confidence [95%CI] = −1.05; −0.26), joint condition (SE = 18.68; 95%CI = −1.16; −0.52) and quadriceps strength (SE = 35.00; 95%CI = 2.53; 17.47). We found that 38.46% and 23.07% of the patients exhibited an improvement in their quadriceps muscle strength and joint condition above the minimum detectable change for both variables (8.21% and 1.79%, respectively). (4) Conclusions: One hundred and eighty degree immersive VR motion visualization can improve the intensity of joint pain in patients with hemophilic knee arthropathy. An intervention using immersive virtual reality can be an effective complementary approach to improve the joint condition and quadriceps strength in these patients.

## 1. Introduction

Hemophilia is a rare disease linked to the X chromosome. It affects 1:10,000 live births. From the pathophysiological point of view, it is characterized by the absence or deficiency of any of the clotting factors. In hemophilia A, clotting factor VIII is missing, while in hemophilia B, the deficiency is related to factor IX [1]. This hematological pathology is characterized by the development of bleeding, mainly in the musculoskeletal system. Intraarticular bleeding (hemarthrosis) is the most common sign, mainly affecting the elbows, ankles, and knees [2].

The recurrence of bleeding events in the same joint causes progressive joint degeneration from an early age [3]; this is known as hemophilic arthropathy [4]. Such arthropathy is characterized by chronic proliferative synovial hypertrophy and osteochondral changes [5]. Joint pain associated with the development of arthropathy can start in childhood. Up to 20% of these patients report chronic pain [6], affecting their perceived quality of life [7].

The gold standard in the treatment of hemophilia for the prevention of hemarthrosis and hemophilic arthropathy is the prophylactic administration of blood clotting concentrates [8] or, more recently, bispecific monoclonal antibodies [9]. Although prophylactic treatment is now widely used, it is scarcely available to patients residing in developing countries. Similarly, most patients now in their adulthood did not have access to such prophylactic treatment during childhood and adolescence [10], thus presenting in advanced degenerative joint damage.

From a neurobiological point of view, pain is a warning system that serves to protect us from potential damage [11]. This system, which receives information from external receptors, also evaluates the relevance of such information in relation to previous information, beliefs, experiences, etc. In the same way, it triggers physiological and behavioral responses [12]. All these responses are influenced by the environment, the tissues, and the evaluation generated by our brain based on all the information, beliefs, emotions, and sensations [12]. This constant assessment of experiences and memories can result in painful responses [13] or may down-modulate the nociceptive information [14].

The therapeutic approach, in connection with pain modulation from the cerebral response and not from peripheral information, has developed a hands-off model in the treatment of pain and, especially, chronic pain [15]. Therapies such as motor imagery, mirror therapy or motion visualization are based on this approach [16]. These therapies are based on the activation of mirror neurons through the observation of a movement. Watching a movement can cause the same cortical activation as if the movement were actually being performed [14]. However, this activation lacks the nociceptive input that could be generated by performing the movement. This makes the movement less relevant [12].

Similarly, the greater the immersion and the reality perceived by the brain, the greater the effect. Therefore, immersive virtual reality (VR) from a first-person perspective can help the patient to feel part of the immersive experience, being a valid option in the approach to patients with pain [17,18].

The aim of this study was to evaluate the effectiveness of an immersive virtual reality intervention in improving the intensity of joint pain, the joint condition, muscle strength, and the range of motion in adult patients with hemophilic knee arthropathy.

## 2. Materials and Methods

### 2.1. Study Design

A prospective, multicenter pilot study was developed in adult patients with hemophilic knee arthropathy. The aim of this pilot study was to evaluate the changes after an immersive virtual reality intervention.

### 2.2. Patient Recruitment and Selection

Patients with hemophilia were recruited in September 2021 from the Hemophilia Associations of Galicia and Malaga and the Spanish Federation of Hemophilia. The study took place between September 2021 and January 2022.

The inclusion criteria of the study were (i) subjects being over 18 years of age; (ii) with a diagnosis of hemophilia A or B; (iii) with a severe hemophilia phenotype (<1% of FVIII/FIX); (iv) with a medical diagnosis of hemophilic knee arthropathy (and more than 4 points on the Hemophilia Joint Health Score) [19]; (v) patients on prophylactic treatment; and (vi) who signed the informed consent document. Patients excluded from the study were those: (i) who developed hemarthrosis during the study period; (ii) without chronic knee pain for at least one year prior to the study; (iii) having neurological or cognitive alterations that prevented their understanding of the questionnaires and evaluation tests; (iv) amputees, epileptic patients, or those with severe vision problems that made it difficult for them to visualize movement with the mobile application; (v) patients who had developed antibodies to clotting factor concentrates (inhibitors); and (vi) those patients who were receiving other physiotherapy treatment at the time of the study.

### 2.3. Ethical Considerations

The main researcher informed the patients about the potential risks and benefits of the study. Subsequently, the patients received an information sheet listing all the characteristics of the study. All subjects signed the informed consent document before being included in the study. The study was conducted in accordance with the Declaration of Helsinki. The study was approved by the Clinical Research Ethics Committee of the Virgen de la Arrixaca University Hospital (ID: 2020-2-9-HCUVA). Prior to the recruitment of patients, the research project was registered (www.clinicaltrials.gov; ID: NCT04549402).

### 2.4. Measurement Instruments

Prior to the experimental phase, the main anthropometric (weight and height) and clinical variables (type of treatment, development of inhibitors, and knee joint condition) of the patients recruited in the study were collected.

Two evaluations were performed: pretreatment (T0) and at the end of the intervention (T1). The primary variable was the intensity of the knee joint pain. The knee joint condition, quadriceps and hamstring muscle strength, and range of motion were the secondary variables. All assessments were performed by the same physiotherapist, with years of experience in the evaluation and treatment of patients with hemophilia, blinded to the study’s objectives.

The intensity of the perceived pain was evaluated using the visual analog scale [20]. This scale has shown an excellent intraobserver reliability (intraclass correlation coefficient [ICC]: 0.97) in assessing the intensity of knee pain [21]. This tool assesses the intensity of pain perceived by patients on a 10 cm line. The patients made a mark on the line that represented the average intensity of their usual joint pain suffered during the last week. Scores ranged from 0 (no pain) to 10 (the worst perceived pain) points.

The joint condition was evaluated using the Hemophilia Joint Health Score [19]. This scale, specific for use in patients with hemophilia, evaluates eight items: swelling and the duration of swelling, pain, atrophy and muscle strength, crepitus, and a loss of flexion and extension. This instrument has shown a high intraobserver reliability (Chronbach’s α = 0.88) in the evaluation of the joint condition in adult patients with hemophilia [5]. The scores, per joint, range from 0 (no joint damage) to 20 points (maximum joint damage).

Muscle strength was measured with a pressure dynamometer (Lafayette Manual Muscle Tester 01165) [22]. Pressure dynamometry has shown high intra-evaluator reliability in adult subjects in knee flexion (CHF: 0.91–0.93) and extension (CHF: 0.82–0.93) movements [23]. The evaluation of quadriceps muscle strength was performed according to the protocol described by Skou et al. [24]. Based on the functional characteristics of these patients, adaptations were made for the evaluation of patients with severe ROM restrictions [2]. With the patient in the supine position and at 75° of hip and knee flexion, the pressure dynamometer was placed perpendicular to the leg, just above the lateral malleolus. The patient was asked to keep the leg in the same position. For the evaluation of hamstring muscle strength, the patient was placed in a prone position and the knee flexed 45°, placing the dynamometer on the back of the leg at the Achilles tendon [25]. For the evaluation of both muscles, the patient was asked to exert two maximum isometric contractions against the dynamometer. These contractions lasted for 5 s, with a 30 s break in between [26]. The mean value of both the measurements was used [27]. The higher the value, the greater the muscle strength. The unit of measurement was Newton.

Knee ROM was assessed with an analog goniometer [28]. This instrument has shown an excellent intraobserver reliability (ICC = 0.91–0.99) in the measurement of mobility in this joint [29]. It was measured in the sagittal plane under no-load pain-free conditions, with the patient in the supine position. The goniometer was positioned with its axis on the joint interline, the reference points being the longitudinal axis of the femur and fibula [30]. The higher the degrees, the greater the range of motion. The unit of measurement is the degree.

Before starting the study, a pilot study was carried out to calculate the evaluator’s intraobserver reliability. Reliability in assessing the joint condition, muscle strength, and range of motion was assessed. Six patients with hemophilia, not included in the study, were evaluated on two consecutive days. An excellent intraobserver reliability was obtained in the variables joint status (CHF = 0.982), and the muscle strength in the quadriceps (CHF = 0.903) and hamstrings (CHF = 0.978), and was good for the range of motion in flexion (CHF = 0.790) and extension (CHF = 0.876) movements.

### 2.5. Intervention

The intervention consisted of immersively visualizing the knee flexion–extension movement. For this purpose, a 180-degree immersive video in a first-person perspective was used. This video was viewed on the patient’s smartphone, regardless of the operating system. In order to view the video immersively, the smartphone was coupled to virtual reality glasses (3D virtual reality glasses with remote control; model Q-MAX) [18]. The video was hosted on YouTube^®^ with access from the He-Mirror App^®^, designed for this study by the research group. After installing the mobile application on the patients’ cell phones, the mobile terminal was coupled to the virtual reality glasses. All patients were given the same model VR glasses, so they all underwent the same intervention with the same program and the same VR system. The patients had to be seated in a chair, with their feet relaxed and only resting on their heels. The intervention was performed for 28 consecutive days at home. The patients performed one daily session. Each session was 15 min long, uninterrupted, without any breaks. During each session, patients had to only watch the movement of both knees on the video, without imagining the movement or performing it. As the procedure was performed in a seated no-load and no-movement position, the patients were informed that even in the event of joint bleeding, they could continue with the intervention. The main study researcher regularly followed up on patients via telephone, clarifying possible doubts about the intervention or solving issues that may arise, encouraging the patients to persevere and adhere to the treatment. Figure 1 shows the intervention as performed by one of the patients included in the study.

### 2.6. Sample Size

The sample size was calculated using the statistical package G*Power (version 3.1.9.2; Heinrich-Heine-Universität Düsseldorf, Germany) before recruiting the patients. Assuming a large effect size (d = 0.80), with an alpha level (type I error) of 0.05 and a statistical power of 80% (1 − β = 0.80), a sample size of 12 patients was estimated. Accounting for potential dropouts during the experimental phase, a total of 13 patients with hemophilia and knee arthropathy were recruited.

### 2.7. Statistical Analysis

The statistical analysis was performed with the software SPPS, version 21.0 for Windows (IBM Company, Armonk, NY, USA). The descriptive statistics (median and interquartile range) of the patients were calculated at the baseline. The changes between the pre- and post-treatment evaluations were calculated with the non-parametric Wilcoxon test. The minimum detectable change (MDC) was calculated with the standard error of measurement (SEM). The SEM was calculated with the formula: SEM = SDpre ∗ √1-intraclass correlation coefficient (ICC) [31]. Based on the SEM, the MDC was obtained (MDC = Z-score ∗ √ 2 ∗ SEM). The confidence level was set at 95% (Z score = 1.96) [32]. In the same way, the proportion of patients whose change after the intervention exceeded the MDC in the study variables was calculated. In this study, an analysis by intent to treat has been carried out. The selected significance level was 0.025 (α = 0.05/2).

## 3. Results

None of the patients developed knee hemarthrosis during the experimental phase as a result of the intervention. There were no adverse effects resulting from the intervention of this study. The median age of the patients was 37 (IR: 14.5) years with a median body mass index of 26.76 (IR: 6.74) kg/m^2^. The majority of patients had a diagnosis of hemophilia A (92.3%). All patients presented a severe hemophilia phenotype (<1% FVIII/FIX) and received prophylactic treatment. Table 1 shows the descriptive characteristics of the patients included in the study.

When comparing the changes after the intervention period, statistically significant differences were observed in the variables for the intensity of the joint pain (Standard error [SE] = 19.31; 95% confidence interval [95%CI] = −1.05; −0.26; *p* < 0.001), joint condition (SE = 18.68; 95%CI = −1.16; −0.52; *p* < 0.001), and quadriceps strength (SE = 35.00; 95%CI = −1.16; −0.52; *p* < 0.001). 95% CI = 2.53; 17.47; *p* = 0.012). Table 2 shows the changes after the study period in each variable.

After the intervention, 38.46% of the patients exhibited an improvement greater than the minimum detectable change (8.21) calculated for quadriceps muscle strength (T0: 235.02; T1: 245.03). Changes in the joint condition (T0: 10.77; T1: 9.92) were greater than the minimum detectable change (1.79) in 23.07% of the patients included in the study. Table 3 shows the calculation of the minimum detectable change and the percentage of patients whose changes exceeded this value.

## 4. Discussion

The aim of this study was to evaluate the changes in the pain intensity, joint condition, range of motion, and muscle strength in patients with hemophilic knee arthropathy after an immersive virtual reality intervention. After the intervention, we found improvements in the perceived pain intensity, joint condition, and quadriceps muscle strength. During the immersive virtual reality intervention, no patient included in the study developed knee hemarthrosis.

Jin et al. [33] noted a significant decrease in pain intensity in patients with total knee arthroplasty after a VR intervention. Byra et al. [34] reported the suitability of using VR in patients with knee and hip osteoarthritis for an effective pain management. Such improvements are due to the multidimensionality of the pain [34]. The illusory effect caused by visualization makes it easier for the brain to evaluate information as something non-aversive, improving downward modulation [12]. These results would be in line with the reduced pain intensity noted in our study.

Adult patients with hemophilia, such as those recruited in this study, have a wide experience of pain from their early childhood as a result of recurring hemarthrosis and arthropathy. It has been described that a change in pain intensity must represent at least two points on the visual analog scale to be clinically relevant [35]. According to our study, the MDC in the pain intensity was 2.027 points and only 7.69% of the subjects experienced changes beyond this value. However, it should be noted that the average intensity of knee pain at the baseline (1.41 points) did not reach two points, so this value should be taken with caution.

Villafañe et al. [36] observed an increase in the knee ROM in patients subject to VR intervention after total arthroplasty. Similarly, Calatayud et al. [37] found changes in the range of shoulder mobility in healthy subjects after a VR exposure with altered visual feedback with regard to the avatar. Changes reported in this study for knee mobility may also be due to the ability to alter statesthesia based on illusory visual inputs [37]. On the other hand, Hsieh et al. [38] found the activation of the same cortical areas after observing a movement or performing it with the hand in healthy subjects. This could cause motion visualization to activate these areas without triggering nociceptive inputs. This non-nociceptive activation may force the brain to reevaluate its available information and modulate the individual’s responses to that movement [12]. These responses can be protective, such as reducing the joint range. Although we found no statistically significant differences in the knee range of motion in our study, these results should be taken with caution considering these two aspects: on the one hand, the small sample size, and on the other, the percentage of subjects (23.07%) who achieved an improvement greater than the minimum detectable change (3.131 degrees) in the loss of a knee extension, which is the most limited movement in this population.

Lee et al. [39] reported improvements in the strength of patients with knee osteoarthritis subject to motion visualization. Although a recent study [2] disclosed no immediate changes in strength improvement after a knee flexion–extension movement visualization session in patients with hemophilic arthropathy, the authors noted a large effect size for the activation of the rectus anterior of the quadriceps. The improved strength of the knee muscles predicted with the electromyographic measurement [40] is confirmed by the changes observed in our study regarding the quadriceps strength. However, caution should be exercised pending randomized clinical studies that confirm these changes.

According to the findings of this study, we are optimistic about the suitability of this intervention in the therapeutic approach to patients with hemophilia. Its easy implementation, low cost, and daily home use promotes the democratization of this protocol, making it more accessible.

### Limitations of the Study

This pilot study has certain limitations that must be considered. On the one hand, the small sample size limits the generalization of results, although there are a series of changes that must be considered. Multicenter randomized clinical studies with an adequate sample size could confirm the results reported in this study. Another limitation is that in this study, the intake of analgesic drugs was not measured and this may affect the intensity of the pain perceived by these patients. In the same way, changes in the functionality of these patients as a result of the intervention have not been evaluated. The evaluation of variables such as functionality, modifications in muscle contraction, and psychosocial variables would provide more information about this intervention and its usefulness in the approach to patients with hemophilic knee arthropathy.

## 5. Conclusions

One hundred and eighty degree immersive VR motion visualization can improve the intensity of joint pain in patients with hemophilic knee arthropathy. Conducting daily immersive motion visualization sessions for 4 weeks can improve the joint condition and quadriceps muscle strength in patients with knee arthropathy. Randomized clinical trials with a larger sample size are needed to confirm the changes observed in this pilot study.

## Figures and Tables

**Figure 1 jcm-11-06216-f001:**
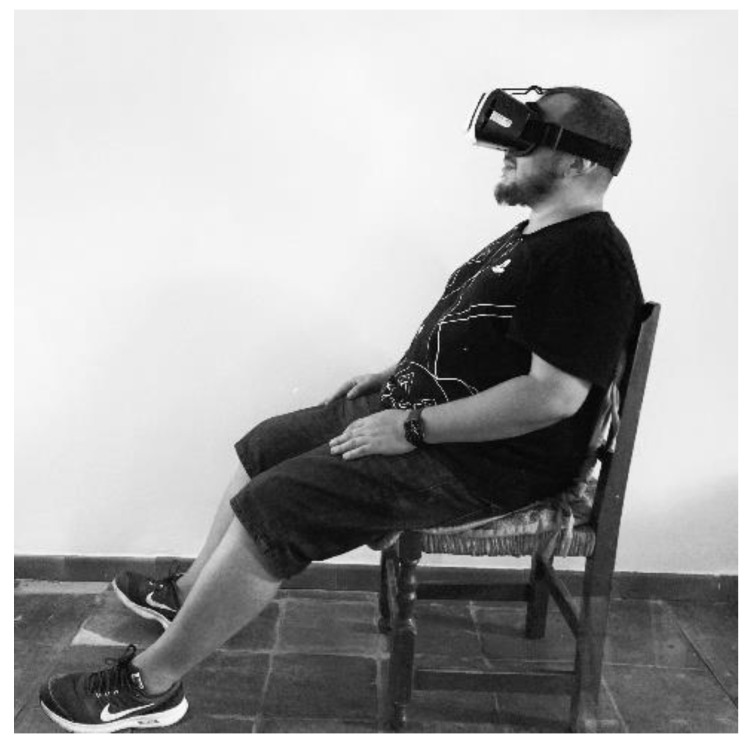
Patient completing the 180° immersive VR motion visualization intervention.

**Table 1 jcm-11-06216-t001:** Descriptive characteristics of patients with hemophilia at baseline.

Variables		Median (IR)
Age (years)		37 (14.5)
Weight (kg)		82.5 (21.2)
Height (cm)		173.0 (8.00)
Body Mass Index (kg/m^2^)		26.76 (6.74)
		n (%)
Type of hemophilia	A	12 (92.3)
	B	1 (7.7)

IR: interquartile range.

**Table 2 jcm-11-06216-t002:** Means (standard deviations) and changes evaluated in the different assessments.

Variables	T0	T1	MD (SE)	95%CI	Sig.
Intensity of joint pain (0–10)	1.41 (1.54)	0.75 (1.40)	−0.66 (19.31)	−1.05; −0.26	0.000
Joint health (0–20)	10.77 (3.44)	9.92 (3.07)	−0.84 (18.68)	−1.16; −0.52	0.000
Flexion (degrees)	114.42 (18.29)	115.04 (18.37)	0.61 (22.79)	−0.41; 1.64	0.254
Loss of extension (degrees)	10.31 (13.46)	9.35 (12.32)	−0.96 (15.85)	−1.94; 0.02	0.063
Quadriceps strength (N)	235.02 (77.07)	245.03 (83.25)	10.01 (35.00)	2.53; 17.47	0.012
Hamstring strength (N)	218.06 (37.67)	220.06 (44.32)	1.99 (35.00)	−7.08; 11.08	0.511

Outcome measures at baseline (T0) and after the 4-week period of interventions (T1); MD: means difference; SE: standard error; 95%CI: 95% interval confidence; Sig.: significance.

**Table 3 jcm-11-06216-t003:** Minimal detectable change of joint status, joint pain, range of motion, and hamstring flexibility evaluated in the different assessments.

Variables	ICC	SEM	MDC (MDCp)
Intensity of joint pain	0.879	0.535	2.027 (7.69)
Joint health	0.985	0.421	1.798 (23.07)
Flexion	0.995	1.293	3.151 (19.23)
Loss of extension	0.991	1.276	3.131 (23.07)
Quadriceps strength	0.987	8.787	8.216 (38.46)
Hamstring strength	0.919	10.721	9.075 (26.92)

ICC: intraclass correlation coefficient; SEM: standard error of measurement; MDC: minimal detectable change; MDCp: proportion of minimal detectable change.

## Data Availability

The data that support the findings of this study are available on request from the corresponding author. The data are not publicly available due to privacy or ethical restrictions.

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
