# Peer review of "Approach to Knee Arthropathy through 180-Degree Immersive VR Movement Visualization in Adult Patients with Severe Hemophilia: A Pilot Study"

_jcm, 2022, doi:10.3390/jcm11206216_

Round 1
Reviewer 1 Report
The authors aimed to evaluate the effectiveness of an immersive virtual reality intervention in improving the intensity of joint pain, joint condition, muscle strength, and range of motion in adult patients with hemophilic knee arthropathy. This is novel research for the management of haemophiliac arthrophathy, and also highlights some important findings.
The work is understandable, but it does not provide detailed information on methods, including procedures, quality of the monitor, and important characteristics of haemophilia patients.
Other comments
1. Introduction: The only effective treatment for the prevention of hemarthrosis and hemophilic arthropathy is the prophylactic administration of blood clotting concentrates: It is not real. Non-factor therapy is widely used for preventing bleeding in the haemophilia population.
2. Materials and Methods line 88: the description of hemophilic knee arthropathy for the subjects was too rough. It was the most important characteristic of the enrolees and needed to be more clearly stated.
3. Materials and Methods line 95 v) patients with antibodies: recommend to change as patients with inhibitors.
4. Materials and Methods line 159. The intervention was performed for 28 consecutive days at home. Each session was 15 minutes: How many sections were done per day? Did patients take any medication, such as analgesics or COXII before the medication? Who inspected the quality of rehabilitation by patients themselves? Was there any pause during the rehabilitation? Were there any instructions for patients on when to stop the rehabilitation, especially when bleeding occurred?
5. Materials and Methods: how did patients do prophylactic therapy before the procedure? Since the patients did rehabilitation every day, what was the trough level of factor VIII or IX?
6. Results: the characteristics of haemophilia should be more detailed, for example, target joints, Hemophilia Activities List, and prophylactic regimens.
7. Results: Any adverse effects must be reported in the result section.
Author Response
Reviewer 1
The authors aimed to evaluate the effectiveness of an immersive virtual reality intervention in improving the intensity of joint pain, joint condition, muscle strength, and range of motion in adult patients with hemophilic knee arthropathy. This is novel research for the management of haemophiliac arthrophathy, and also highlights some important findings.
The work is understandable, but it does not provide detailed information on methods, including procedures, quality of the monitor, and important characteristics of haemophilia patients. Dear reviewer. We have tried to answer all the doubts raised, hoping that the changes are in accordance with your questions
Other comments
- Introduction: The only effective treatment for the prevention of hemarthrosis and hemophilic arthropathy is the prophylactic administration of blood clotting concentrates: It is not real. Non-factor therapy is widely used for preventing bleeding in the haemophilia population. The treatment defined as the gold standard in the approach to patients with hemophilia is the prophylactic regimen. Given the appearance of new drugs in recent years, we have included a reference to these monoclonal antibodies.
- Materials and Methods line 88: the description of hemophilic knee arthropathy for the subjects was too rough. It was the most important characteristic of the enrolees and needed to be more clearly stated. It has been indicated that one of the inclusion criteria is the medical diagnosis of hemophilic arthropathy and having more than 4 points on the HJHS scale. Hemophilic arthropathy was already described in the Introduction
- Materials and Methods line 95 v) patients with antibodies: recommend to change as patients with inhibitors. As the reviewer points out, the term inhibitor has been included to avoid confusion for the reader.
- Materials and Methods line 159. The intervention was performed for 28 consecutive days at home. Each session was 15 minutes: How many sections were done per day? It has been indicated in the text that the patients performed a daily session. Did patients take any medication, such as analgesics or COXII before the medication? As indicated in the limitations of the study, one of the main limitations of this study was the non-evaluation of the analgesic drugs used by the patients. Although analgesic control is controlled by his hematologist, this variable was not controlled. Who inspected the quality of rehabilitation by patients themselves? Before the start of the study, a mobile app (He-Mirror) was designed and installed on the patients' mobile phones. The mobile terminal was coupled to the virtual reality glasses that were given to all patients so that in this way, all patients received the same intervention, with the same program and the same virtual reality system. The main researcher of the study carried out a periodic telephone follow-up answering possible doubts about the intervention or incidents in carrying out the protocol, encouraging the patients to continue adhering to the treatment. Was there any pause during the rehabilitation? Were there any instructions for patients on when to stop the rehabilitation, especially when bleeding occurred? All the sessions were carried out uninterruptedly, without breaks, during the 15-minute duration. In the event of joint bleeding, as the intervention was performed in a sitting position, without weight and without movement, patients were told that they could continue with the treatment program.
- Materials and Methods: how did patients do prophylactic therapy before the procedure? Since the patients did rehabilitation every day, what was the trough level of factor VIII or IX? The patients were in prophylactic treatment, continuing with the same dosage as that prescribed by their hematologist. The intervention did not require active movement of the patient, ambulation or load, so there was no risk associated with the intervention (beyond possible trauma due to having the glasses and not being able to see the surroundings). Patients were instructed at the start of the study not to change their lifestyle, dosage, or periodization of prophylactic treatment. The trough, minimum or peak pharmacokinetic level were not evaluated in the study since these medical data were only handled by the reference hematologist. Likewise, since it was not the object of study and based on the characteristics of the procedure methodology, it was not a determining factor to control.
- Results: the characteristics of haemophilia should be more detailed, for example, target joints, Hemophilia Activities List, and prophylactic regimens. All patients with hemophilia included in the study had to be on prophylactic treatment and had severe hemophilia (selection criteria). For this reason, this data is not repeated in the Results section because it is a mandatory criterion. The HAL (Hemophilia Activities List) questionnaire was not administered to the patients and therefore has not been included in the study results. Table 2 (in the column of pre-treatment values) shows the main characteristics of the affected joint: joint status (Hemophilia Joint Health Score), joint pain, range of motion)
- Results: Any adverse effects must be reported in the result section. Results: Any adverse effects must be reported in the result section. Information on adverse effects in the study has been included at the beginning of the Results
Reviewer 2 Report
This is a very interesting study about the use of motor imagery in patients with haemophilia.
Topic is novel and interesting, and the article is well written
Some minor suggestions:
- about the possible application of motor imagery, I'd suggest the following reference (https://www.mdpi.com/2411-5142/5/4/89)
- I'd add some informations more about type of the study
- methodology is very strong, I really appreciated it
- I suggest to add some practical implications of your study
Author Response
Reviewer 2
This is a very interesting study about the use of motor imagery in patients with haemophilia. Topic is novel and interesting, and the article is well written
Some minor suggestions:
- About the possible application of motor imagery, I'd suggest the following reference (https://www.mdpi.com/2411-5142/5/4/89) This reference has three problems to be considered for this paper. Firstly, the subjects included in the survey were healthy military people. Secondly, the aim was to know the modification of the response time to visual and auditive stimulation, very different to our purpose. Thirdly the authors mixed in their protocol observation and imagination doing that the results couldn’t be differenced between the effects of each intervention. These problems make it impossible to compare with our work
- I'd add some informations more about type of the study. The type of patients included in the study design and the purpose of this study have been included.
- Methodology is very strong, I really appreciated it. Thank you very much for your comments. Your feedback is appreciated.
- I suggest to add some practical implications of your study. Practical implications have been included before the Limitations of the study
- Pay attention to english grammar and punctuation, all over the manuscript - pay attention to acronyms, also in the description of tables and figures - use short periods, to better focus the attention of the readers - use paragraphs to explain better your concepts. The text and grammar have been reviewed by a qualified native translator who has reread and modified those sections that required grammatical nuances.
Round 2
Reviewer 1 Report
The authors have already addressed my comments.